# Complete-Arch Accuracy of Four Intraoral Scanners: An In Vitro Study

**DOI:** 10.3390/healthcare9030246

**Published:** 2021-03-01

**Authors:** Giordano Celeghin, Giulio Franceschetti, Nicola Mobilio, Alberto Fasiol, Santo Catapano, Massimo Corsalini, Francesco Grande

**Affiliations:** 1Department of Prosthodontics, Dental School, University of Ferrara, 44123 Ferrara, Italy; frngli1@unife.it (G.F.); mblncl@unife.it (N.M.); alberto.fasiol@unife.it (A.F.); francesco.grande90@gmail.com (F.G.); 2Interdisciplinary Department of Medicine (DIM)–Section of Dentistry, “Aldo Moro” University of Bari, 70124 Bari, Italy; massimo.corsalini@uniba.it

**Keywords:** intraoral scanner, digital impressions, complete dental arch accuracy

## Abstract

The purpose of this study is to define the accuracy of four intraoral scanners (IOS) through the analysis of digital impressions of a complete dental arch model. Eight metal inserts were placed on the model as reference points and then it was scanned with a laboratory scanner in order to obtain the reference model. Subsequently, the reference model was scanned with four IOS (Carestream 3600, CEREC Omnicam, True Definition Scanner, Trios 3Shape). Linear measurements were traced on an STL file between the chosen reference points and divided into four categories: three-element mesiodistal, five-element mesiodistal, diagonal, and contralateral measurements. The digital reference values for the measurements were then compared with the values obtained from the scans to analyze the accuracy of the IOS using ANOVA. There were no statistically significant differences between the measurements of the digital scans obtained with the four IOS systems for any of the measurement groups tested.

## 1. Introduction

One important option in the production of dental prostheses nowadays is the use of CAD/CAM technologies [1,2]. Conventional impressions may be combined with CAD/CAM technologies through digitalization of the impressions or of the final casts, achieved with the use of laboratory scanners [3]. Intraoral scanners (IOS) allow one to create, during impression taking, an STL file that is directly available to the technician. Certain limits of the conventional workflows can be overcome, such as errors in impression taking, plaster model production, storage of materials, and possible risks of cross-infection between the dental office and lab [4,5,6]. IOS allow a real-time check of the impression results, possibly avoiding additional appointments, new impressions, and further inconvenience for the patient [7]. In addition, the possibility of complete digitalization of the CAD/CAM process guarantees direct access to subtractive and additive technologies [3], which allow the processing of materials in monolithic form [8].

However, the accuracy and precision of the impressions are equally important for IOS in order to accurately reproduce a patient’s anatomy. Several studies [5,7,9] have shown the equivalent accuracy of IOS in impression taking as compared to conventional impression materials for various prosthetic restorations, such as single crowns and short-span restorations of up to five elements [10], with various different materials being used (metals, resins, ceramics) [11]. However, for some clinical situations, such as edentulous patients [12], subgingival preparations [13], and full-arch impressions [7,14], IOS have not been perfectly coded yet. In fact, they could have difficulties in detecting deep marginal lines, because light cannot detect soft tissues and it may be more difficult to read the entire finishing line [15]. The use of IOS for partial and complete dental prostheses is still questionable due to problems related to registering soft tissue dynamics and the absence of reference points [16,17,18].

However, the use of IOS in complete arches still needs to be correctly defined; the literature does not support the use of IOS for full-arch impressions in prosthetic dentistry due to the challenging nature of these cases [19], even if their use for full-arch impressions for orthodontic treatments is accepted [7]. The aim of the present study was to evaluate the levels of accuracy, in terms of trueness, of four different IOS used to make impressions of an anatomical model of a complete maxillary arch.

## 2. Materials and Methods

An anatomical model (ANA-4 Frasaco, Tettnang, Germany) of a complete upper dental arch was used: eight spheres of polyethylene measuring 3 mm in diameter were placed on the dental elements using ethyl cyanoacrylate. The spheres were characterized by the presence of a cylindrical metal insert with a diameter of 0.61 mm and heights ranging between 0.34 and 0.88 mm, with the most coronal portion characterized by an irregular surface, dictated by the cut made with a tungsten carbide disc. The spheres were positioned on the model at these points: distal fossa of the occlusal surface of the right third molar (h 0.84 mm); central fossa of the occlusal surface of the right first molar (h 0.38 mm); mesial fossa of the occlusal surface of the right first premolar (h 0.41 mm); palatine face of the right central incisor (h 0.51 mm); palatine face of the left central incisor (h 0.35 mm); mesial fossa of the occlusal surface of the left first premolar (h 0.76 mm); central fossa of the occlusal surface of the left first molar (h 0.88 mm); distal fossa of the occlusal surface of the left third molar (h 0.85 mm) (Figure 1). The anatomical model was scanned using a Swing HD laboratory scanner (Ascoli Piceno, Italy) to create a STL file to be used as a reference model, with an accuracy of 10 µm, as declared by the manufacturer according to ISO 12836:2015. Four STL files were then derived by scanning the master model using four different IOS: a Carestream 3600 (Carestream Health, version 3.1.0; Rochester, NY, USA), CEREC Omnicam (Dentsply Sirona, version 4.6.1; York, PA, USA), True Definition Scanner (3M; Saint Paul, MN, USA), and Trios 3Shape (version 1.18.2.6; Copenaghen, Denmark). From each STL file, a three-dimensional model was created using the “model creator” module in DentalCAD 2.2 Valletta software (Exocad GmbH, Darmstadt, Germany). The five STL files (one reference model and four IOS models) were subsequently analyzed using DentalCAD 2.2 Valletta dental software in order to identify the most coronal point of each individual metal insert of the spheres in the network of points formed by the vertices of the triangles of the STL language (Figure 2). The identified points were used as references to trace 28 linear measurements (mm) using the “virtual ruler” function in DentalCAD 2.2 Valletta, with a resolution of 0.001 mm (Figure 3). The 28 measurements were determined by the reciprocal connections of all 8 landmarks positioned on the dental model and were divided into four categories, as shown in Table 1. 

Although the distance between the right central incisor and the left central incisor does not provide the interposition of a third element between the two reference elements, it falls into the mesio-distal (3 elements) category, since it represents the shortest possible distance between the landmarks. The measurements were carried out in order to simulate the extensions of different types of rehabilitation, starting from three-tooth short-span restoration up to full-arch rehabilitations. The same operator performed the scanning and digital measurements. The measurements obtained from the reference model were compared to the measurements obtained from the four IOS models. The difference Δ, obtained for each individual measurement by the formula (A (reference measurement) –A^1^ (IOS STL model measurement) = Δ), represents the value in mm of the distortion (positive or negative) of the IOS measurement compared to digital reference model. The Kolmogorov–Smirnov test was performed to test the normal distribution of the data and one-way analysis of variance (ANOVA) was performed to test the differences between measurements. SPSS software v.22 for MAC OS X was used. The *p* value was set at 0.05.

## 3. Results

The ANOVA did not show statistically significant differences between the measurements of the digital scans obtained with IOS systems for any of the groups tested: mesio-distal measurements (3 elements) = F (3.24) = 1.192; *p* = 0.334; mesio-distal measurements (5 elements) = F (3.20) = 0.336; *p* = 0.799; contralateral measurements = F (3.8) = 0.843; *p* = 0.508; diagonal measurements = F (3.44) = 0.282; *p* = 0.838. Table 2 shows the mean values (expressed in µm) of the comparison of the linear measurements carried out on the STL files of the digital models created from the impressions of the Frasaco model obtained using the Swing HD (DOF), Carestream 3600, CEREC Omnicam, Trios 3Shape, and 3M True Definition Scanner. The overall accuracy data for the complete arch are as follows: 63 ± 48 µm for CS 3600; 63 ± 53 µm for CEREC Omnicam; 50 ± 42 µm for 3M True Definition Scanner; 55 ± 42 µm for Trios 3Shape.

## 4. Discussion

Impression taking is a crucial phase in prosthodontics—the accuracy of this process affects the quality of the final prosthesis, and in the long term its survival [5]. Impressions obtained with conventional materials (polyether and polyvinylsiloxane) are the most common procedures in clinical practice; conventional workflows and CAD/CAM technologies can be combined through indirect digitalization [20]. However, new digital techniques allow the full digitalization of the workflow—intraoral digital scanners, through direct scanning, can create a virtual 3D model [21,22]. The main feature to be evaluated in an intraoral scanner is accuracy, which is the result of the analysis of trueness and precision [23]. We define trueness as the closeness of agreement between a measurement result and a real value [14]. In other studies, the accuracy of digital impressions is similar to that of conventional impressions for single-tooth restorations and fixed partial prostheses of up to 4 elements [7,19,23,24]. Digital impressions do not seem to have the same accuracy as conventional impressions for partial fixed restorations with more than 5 elements or prostheses involving complete arches with both natural teeth and implants. However, the literature does not support the use of IOS for full-arch impressions yet [12,25,26,27,28,29]. The aim of this study was to evaluate the level of accuracy, in terms of trueness, of four different intraoral scanners used for the impression of an anatomical model of a complete maxillary arch. In vitro studies have used different procedures to investigate the accuracy of full-arch digital models—the most used procedure involves a three-dimensional analysis generated by superimposing the digital model with a reference model using best-fit algorithms and by calculating the mean differences of the surface areas. Authors in the literature avoided scan superimposition because of errors caused by superimposition computing process, especially those in larger data sets, such as full-arch scans, and for high deviations between superimposed areas [30]. Other studies in the literature used geometrical forms, which were verified with a coordinate-measuring machine (CMM) [31]; even if they show high trueness, these machines acquire just a small number of points from the refence model surface, and the shape of the surface must be known before scanning in order to achieve a precise model with a CMM. Furthermore, the tip of the tactile probe has a certain diameter, which means small morphological structures cannot be detected. On the other hand, with a reference scanner it is possible to acquire dental surface information with no prior knowledge of the morphology. With an accuracy of up to 10 micron, reference scanners are useful below the average deviations of conventional full impressions, and therefore are suitable for accuracy measurements [23]. In this study, a laboratory scanner was used to obtain both the reference data and the differences from the sample data derived using intraoral scanners. Although the use of a reference scanner is a limiting factor in this study (the maximum error margin is 10 μm), previous studies in the literature have stated that the use of a reference scanner for the evaluation of trueness is possible [23]. For each IOS system, the values of distortion of the three-element mesio-distal measurements were lower than the values for the other measurement categories, while the values for the five-element mesio-distal measurements were lower than the values of the contralateral and diagonal measurement categories. The analyzed IOS systems showed similar deviation models, with the lowest trueness values obtained for the contralateral linear measurements category (except for the True Definition Scanner 3M, for which the lowest values were for the diagonal linear measurements category). It can be concluded that as the scan area increases, the amount of distortion of the impression increases as well. The results obtained from the analysis of trueness show that there are no statistically significant differences between the four different intraoral scanners. The limits of this study are the fact that it is an in vitro analysis, which is not able to represent the different clinical realities that can present themselves; therefore, there were no clinical variables, such as the presence of blood and saliva, and the technical difficulty was dictated by the limited oral cavity space. The analysis did not take into account the three-dimensionality of the dental surfaces—linear measurements do not allow evaluation of the accuracy of a digital impression in terms of the angulation and curvature of the dental walls, aspects that play fundamental roles in the morphological reading of dental preparations. Finally, not all IOS that are currently commercially available were used in this study.

## 5. Conclusions

Within the limitations of the current study, the following conclusions may be drawn—there were no statistically significant differences between the four IOS systems in terms of the accuracy of impressions of a complete dental arch, and as the extension of the scanned surface increased, there was an increase in the distortion of the impression. Future developments of intraoral scanners and three-dimensional printing systems could expand the application fields to the clinical use of fully digital workflows, especially in the rehabilitation of complete arches. Further in vitro and in vivo studies are needed to carefully evaluate the progress of digital technologies in dentistry.

## Figures and Tables

**Figure 1 healthcare-09-00246-f001:**
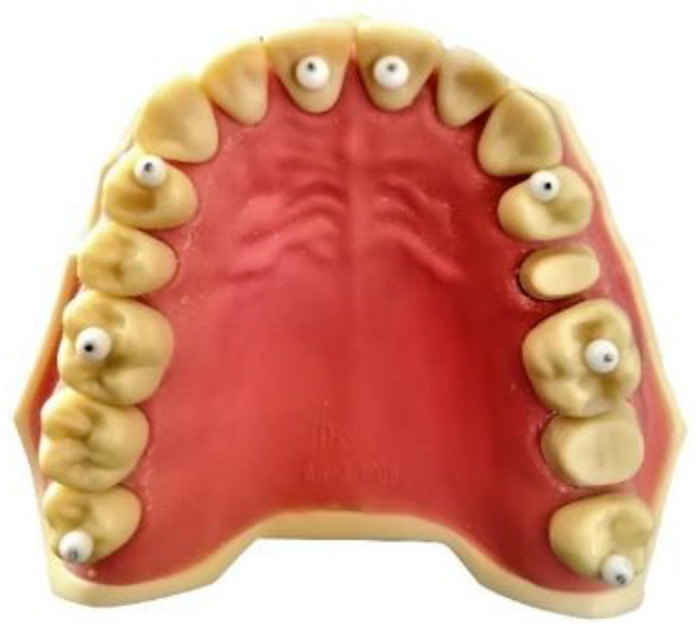
Master model.

**Figure 2 healthcare-09-00246-f002:**
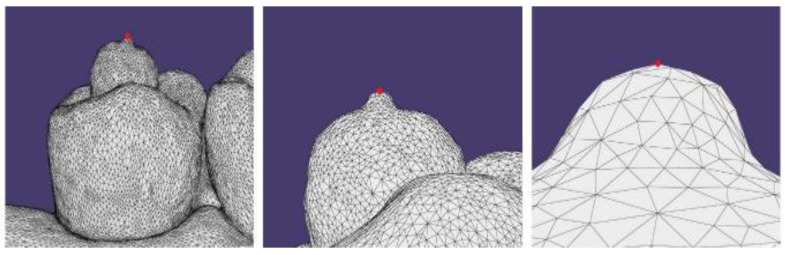
Reference point with increasing magnification.

**Figure 3 healthcare-09-00246-f003:**
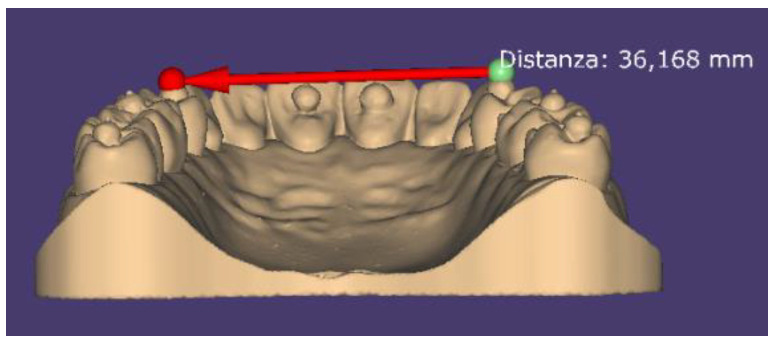
Example of contralateral measurement between premolars.

**Table 1 healthcare-09-00246-t001:** Measurement categories.

Categories	Measurements
Mesio-distal (3 elements)	Right third molar–Right first molar Right first molar–Right first premolar Right first premolar–Right central incisor Left central incisor–Left first premolar Left first premolar–Left first molar Left first molar–Left third molar Right central incisor–Left central incisor
Mesio-distal (5 elements)	Right third molar–Right first premolar Right first molar–Right central incisor Right first premolar–Left central incisor Right central incisor–Left first premolar Left central incisor–Left first molar Left first premolar–Left third molar
Contralateral	Right first premolar–Left first premolar Right first molar–Left first molar Right third molar–Left third molar
Diagonal	Right third molar–Right central incisor Right third molar–Left central incisor Right third molar–Left first premolar Right third molar–Left first molar Right first molar–Left central incisor Right first molar–Left first premolar Right first molar–Left third molar Right first premolar–Left first molar Right first premolar–Left third molar Right central incisor–Left first molar Right central incisor–Left third molar Left central incisor–Left third molar

**Table 2 healthcare-09-00246-t002:** Mean |Δ| values (µm).

Mean |Δ| Values	CS3600	Omnicam	TDS	Trios	*p* Value
Mesio-Distal Measurements (3 elements)	38 ± 18	34 ± 19	22 ± 14	43 ± 32	0.334
Mesio-Distal Measurements (5 elements)	64 ± 59	48 ± 27	43 ± 42	43 ± 33	0.799
Contralateral Measurements	83 ± 69	136 ± 83	58 ± 52	103 ± 47	0.508
Diagonal Measurements	73 ± 51	69 ± 55	67 ± 46	55 ± 47	0.838
Full-Arch Overall Value	63 ± 48	63 ± 53	50 ± 42	55 ± 42	-

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
