# Peer review of "Complete-Arch Accuracy of Four Intraoral Scanners: An In Vitro Study"

_healthcare, 2021, doi:10.3390/healthcare9030246_

Round 1

Reviewer 1 Report

The grammar can be improved to increase the comprehension of the words for readers, any sentence revisions can be performed at the editorial stage to avoid another review cycle for the manuscript.

Author Response

Dear reviewer,

thanks for your comments. We improved the grammar in the article as you suggested.

Reviewer 2 Report

The authors had addressed my concerns to the earlier submission.

Author Response

Dear reviewer,

thank you for your revision

Reviewer 3 Report

Measurements methodology (M&M section) was clarified.

Author Response

Dear reviewer,

thank you for your revision

This manuscript is a resubmission of an earlier submission. The following is a list of the peer review reports and author responses from that submission.

Round 1

Reviewer 1 Report

The topic is within the scope of the journal. The English grammar requires some editing.

Several sentences need to be revised to improve their comprehension for readers. Abbreviations need to be explained. The authors citation style is annoying because they make many opinions without giving a citation, but for some citations they give too many; “(3,7-19)” thirteen is too many. The publication may be accepted if the authors revised it according to the peer-review comments to the satisfaction of the editor.

Title

  1. Acceptable

Abstract

  1. Page 1, line 20: To improve its clarity, remove Page 1: “IOS represent nowadays a solution to overcome limits of conventional workflows”
  2. Page 1, line 21: State what IOS is: “Intraoral scanners.”
  3. Page 1, line 27: Change the sentence: “The digital reference values of the measurements were 27 then compared with the values obtained from the scans to analyze the accuracy of the IOS. The 28 ANOVA performed did not show statistically significant differences between the measurements of 29 the digital scans obtained with the four IOS systems” to: “The digital reference values of the measurements were then compared with the values obtained from the scans to analyze the accuracy of the IOS. There were no statistically significant differences between the measurements of the digital scans obtained with the four IOS systems.
  4. Page 1, line 31: Change “trueness” to “accuracy.”
  5. Page 1, line 34: I don't understand the distortion because there was no significant differences, so how did it arise?

Introduction

  1. The authors citation style is annoying because they make many opinions without giving a citation, but for some citations they give too many; “(3,7-19)” thirteen is too many. Consider distributing citations to support each statement, rather than using them excessively when only 1 or 2 is sufficient.

Materials and Methods

  1. For each supplier and manufacturer name the city and country.

Results

  1. Please edit the sentences, by describing the differences and then the measurements. For example move the first sentence to after the second.

Discussion

  1. There is a lack of citations for many of the statements.

References

  1.  

Figures

  1. Table 2, last column and last line is missing a P value.

Reviewer 2 Report

  1. Having the entire introduction in 1 paragraph is rather amateurish and making it very difficult to read. Same for the rest of the manuscript - each section is one paragraph. I will recommend a complete re-write of this manuscript before a re-review.
  2. These 2 sentences in the abstract are essentially saying the same time - "The 28 ANOVA performed did not show statistically significant differences between the measurements of 29 the digital scans obtained with the four IOS systems, for all the measurements groups tested. There 30 is no statistically significant difference between the four IOS tested in the analysis of the trueness of 31 impressions of a complete dental arch and the comparison between the different measurement 32 categories has shown that...". What is 28 ANOVA? There are 6 steps to any hypothesis testing and the 6th step is interpretation - what the results meant? In the words of high-school chemistry - don't tell me the results of nitrate test or chloride test but tell me whether the compound that I had given to you is Calcium chloride or Calcium nitrate? In this case, are the results from these machines comparable or not?
  3. What does p-value in Table 2 meant? What are you comparing?

Reviewer 3 Report

The manuscript is well-presented and the structure is clear.

M&M section: authors should specify if measurements were taken by different operators (different also from the operator who performed the scanning) and if there was adequate intra-examiner calibration.